# Integrating Habitat Prediction and Risk Assessment to Prioritize Conservation Areas for the Long-Tailed Goral (*Naemorhedus caudatus*)

**DOI:** 10.3390/ani15192848

**Published:** 2025-09-29

**Authors:** Soyeon Park, Minkyung Kim, Sangdon Lee

**Affiliations:** Department of Environmental Science & Engineering, Ewha Womans University, Seoul 03760, Republic of Korea; psy25730@ewha.ac.kr (S.P.); enviecol@ewha.ac.kr (M.K.)

**Keywords:** endangered species, MaxEnt, InVEST, prioritization, conservation strategy

## Abstract

**Simple Summary:**

The long-tailed goral (*Naemorhedus caudatus*) is an endangered species in South Korea, with populations severely declining due to anthropogenic pressures. This study integrates habitat prediction and risk assessment within a spatial prioritization framework to identify areas critical for the effective conservation of long-tailed goral habitats. The results indicate that core areas with both high suitability and ecological value should be prioritized when designating protected areas. These findings emphasize the importance of targeted habitat conservation for endangered species and highlight the need for ongoing research to advance sustainable biodiversity conservation strategies.

**Abstract:**

Human activities have accelerated the extinction of species, driving biodiversity loss and ecosystem degradation. Establishing protected areas (PAs) that encompass habitats of endangered species is essential for achieving biodiversity conservation and ecosystem protection goals. This study aimed to identify and prioritize critical conservation areas for the endangered long-tailed goral (*Naemorhedus caudatus*) in five regions of Gangwon and Gyeongbuk Provinces, South Korea. The MaxEnt model was applied to predict the potential habitat of the species, considering key environmental factors such as topographic, distance-related, vegetation, and land cover variables. The InVEST Habitat Risk Assessment (HRA) model was used to quantitatively assess cumulative risks within the habitat from the impacts of forest development and anthropogenic pressures. Subsequently, the Zonation software was employed for spatial prioritization by integrating the outputs of the models, and core conservation areas (CCAs) with high ecological value were identified through overlap analysis with 1st-grade areas from the Ecological and Nature Map (ENM). Results indicated that suitable habitats for the long-tailed goral were mainly located in forested regions, and areas subjected to multiple stressors faced elevated habitat risk. High-priority areas (HPAs) were primarily forested zones with high habitat suitability. The overlap analysis emphasized the need to implement conservation measures targeting CCAs while also managing additional HPAs outside CCAs, which are not designated as ENM. This study provides a methodological framework and baseline data to support systematic conservation planning for the long-tailed goral, offering practical guidance for future research and policy development.

## 1. Introduction

Rapid industrialization has intensified human activities and placed significant anthropogenic pressures on ecosystems, resulting in environmental changes such as habitat loss and fragmentation [1]. These impacts threaten the survival of species and accelerate extinction rates [2,3]. Endangered species are particularly vulnerable to extinction due to their small population sizes and high ecological plasticity [4]. The decline of endangered species directly contributes to biodiversity loss and adversely affects the quality and stability of ecosystem services that benefit humans [5,6]. As endangered species serve as indicators of ecosystem health, systematic conservation planning based on a comprehensive understanding of their ecological characteristics and distribution is essential [7]. The long-tailed goral (*Naemorhedus caudatus*), an endangered species in South Korea, has experienced a continuous decline in its population due to habitat loss, poaching, and extreme snowfall associated with climate change, which are factors exacerbated by human activities [8,9]. It is classified as Vulnerable (VU) on the International Union for Conservation of Nature (IUCN) Red List and listed in Appendix I of the Convention on International Trade in Endangered Species of Wild Fauna and Flora (CITES) [10,11]. Due to its limited habitat range and small population size, ecological and distributional data for the long-tailed goral remain scarce [8]. Although restoration projects have been initiated for its conservation, ecological and distribution studies remain insufficient [8,12].

Designated protected areas (PAs) that encompass the habitats of endangered species are one of the most effective conservation strategies [13,14]. Such habitats can be identified not only through field surveys but also through predictive approaches such as Species Distribution Models (SDMs) [15,16]. The Maximum Entropy (MaxEnt) model requires only presence records and is widely used in South Korea, where ecological studies mainly rely on field survey data. This approach is particularly effective for endangered species with restricted habitats and limited population data, and has become a crucial tool in various biodiversity conservation studies [17]. However, although the MaxEnt model has been widely applied both domestically and internationally, its use for predicting the distribution of the long-tailed goral has been limited, with most previous studies focusing on Seoraksan National Park, the largest distribution area, or analyzing the species alongside others.

When designating PAs, it is important to minimize or eliminate existing threats within the boundaries of the habitat [2]. Consequently, risk assessments addressing multiple threats should be conducted, as anthropogenic disturbances can compromise species survival and undermine conservation efforts [18]. The Habitat Risk Assessment (HRA) model of the Integrated Valuation of Ecosystem Services and Tradeoffs (InVEST) models enables the quantitative evaluation of potential habitat risks by considering the degree of exposure and cumulative stress from human activities [19]. Although initially developed for marine ecosystems, the HRA model has also proven effective in terrestrial environments [19,20]. Habitat risk assessment methods support the development of management strategies aimed at minimizing impacts on ecosystem services and biodiversity, and the HRA model is expected to be significant in the designation of PAs.

Establishing systematic PAs requires setting conservation priorities that maximize ecological benefits relative to the costs of conservation. Decision-support tools, such as Zonation, Marxan, and C-Plan, have been employed to evaluate conservation value and spatially prioritize areas for protection [13,21]. Analyzing spatial conservation priorities in advance, by considering areas with potential habitat value as well as areas at risk from threat factors, can enhance the efficiency of designating and managing PAs.

This study develops a habitat conservation strategy for the endangered long-tailed goral (*Naemorhedus caudatus*) by integrating species distribution and risk assessment models. The MaxEnt model was applied to predict the potential distribution of the species, and the InVEST HRA model was used to assess cumulative habitat risks. The outputs were integrated using Zonation software (version 2.1) to quantitatively determine spatial conservation priorities within the habitat, providing a scientific basis for identifying optimal conservation areas.

## 2. Materials and Methods

### 2.1. Study Species

The long-tailed goral (*Naemorhedus caudatus*) was selected as the target species (Figure 1). It is highly vulnerable to human activities and is designated as Endangered Wildlife Class I by the Ministry of Environment of South Korea, necessitating urgent habitat conservation measures. The long-tailed goral is a bovid mammal of the order Artiodactyla that retains the appearance of ancestral bovids. It is recognized for its conservation significance and is designated as a Natural Monument in South Korea [8]. The long-tailed goral is restricted to the parts of Korean Peninsula, eastern Russia, and northeastern China [22]. Within South Korea, the long-tailed goral primarily inhabits high-altitude forested areas, notably the Demilitarized Zone (DMZ), Seoraksan Mountain, and the mountainous regions of Gangwon-do (Gangwon) and Gyeongsangbuk-do (Gyeongbuk) Provinces [8,11]. The habitat of the long-tailed goral serves as an indicator of the ecological health of forest ecosystems in these regions and plays a critical role in the designation of protected areas [12].

### 2.2. Study Site

The criteria for selecting the study sites among the main habitats of the long-tailed goral focused on regions where presence records could be readily obtained, and where areas at risk from developmental impacts could be identified. Based on the occurrence records of the long-tailed goral in South Korea, areas with relatively high population density and suitable conditions for assessing habitat disturbances and anthropogenic threats were identified. Five regions were selected as study sites: Samcheok-si (Samcheok) and Taebaek-si (Taebaek) in Gangwon Province, and Bonghwa-gun (Bonghwa), Uljin-gun (Uljin), and Yeongyang-gun (Yeongyang) in Gyeongsangbuk-do (Gyeongbuk) Province (Figure 2). The sites are situated around 37° latitude and 127° longitude in East Asia. The administrative areas cover 1187.84 km^2^ for Samcheok, 303.44 km^2^ for Taebaek, 1202.70 km^2^ for Bonghwa, 990.63 km^2^ for Uljin, and 815.87 km^2^ for Yeongyang, totaling approximately 4500.47 km^2^. As of 2022, the study sites had a high forest cover rate of 83.6%, with agricultural land accounting for 8.12%, primarily concentrated along the outer boundaries of the study areas (Figure 2c). Therefore, the study sites are mostly forested areas, reflecting the habitat preference of the long-tailed goral.

### 2.3. Study Flow

The study was conducted in accordance with the sequential steps outlined in Figure 3. Relevant literature was reviewed, and datasets for the long-tailed goral were compiled. The potential habitat and risk areas was modeled using the MaxEnt (version 3.4.4) and InVEST HRA (version 3.14.2) models. The outputs from these models were then integrated into a spatial prioritization process using Zonation software (version 2.1) to identify high-priority conservation areas within the predicted habitat of the long-tailed goral.

### 2.4. Potential Habitat Prediction

The MaxEnt model predicts the potential distribution of species by applying machine learning techniques to occurrence records and environmental variables [23,24]. Model performance is evaluated using the Area Under the Curve (AUC) of the Receiver Operating Characteristic (ROC) curve, with values closer to 1.0 indicating higher predictive accuracy [25].

Occurrence data for the long-tailed goral were obtained from the 4th and 5th Nationwide Natural Environmental Survey records (2014–2022) conducted by the National Institute of Ecology (NIE). Records containing spatial errors or missing coordinate information were removed. To minimize spatial sampling bias, occurrence points were distributed at 1 km intervals using R Studio (version 4.4.2) (https://posit.co, accessed on 1 March 2024). Therefore, out of a total of 65 occurrence points, 61 points were used as input data for the MaxEnt model.

Environmental variables were selected based on a review of previous studies on the ecology and distribution of the long-tailed goral [26,27,28,29,30,31]. These variables were grouped into four categories: topographic, distance-related, vegetation, and land cover. All datasets were processed at a spatial resolution of 30 m × 30 m using ArcGIS Pro (version 3.1.3). To reduce multicollinearity, variables with high correlation coefficients (|r| > 0.7) were excluded based on Spearman’s rank correlation analysis, resulting in a final selection of 11 environmental variables for the MaxEnt model (Table 1).

The MaxEnt (version 3.4.4) (https://biodiversityinformatics.amnh.org/open_source/maxent, accessed on 1 March 2024) was implemented using the bootstrap method, with 20% of the occurrence points reserved as test data for model validation, and was replicated 15 times. A jackknife test was performed to assess the relative contribution of each environmental variable. Model performance was evaluated using the AUC value. The final logistic output was scaled between 0 and 1, and the maximum training sensitivity plus specificity (MTSS) threshold value was applied to classify the suitability of the habitat into binary categories.

### 2.5. Habitat Risk Assessment

The HRA model in the InVEST models quantifies the cumulative risk (R) of anthropogenic stressors on habitats by integrating two key components, i.e., exposure (E) and consequence (C), based on the spatiotemporal overlap between habitats and stressors [32].

In this study, the suitable areas for the long-tailed goral, as predicted by the MaxEnt model, were defined as the spatial extent for HRA modeling. Habitat resilience was evaluated based on the focal species [33], with evaluation criteria scored on a scale of 1 to 3, derived from a literature review [8,9,10,11,34] (Table 2).

The selection of stressors was based on the primary drivers of the long-tailed goral population decline, focusing specifically on anthropogenic structures and forest development. Five primary stressors were identified: roads, mountain trails, deforestation, quarries, and wind power plants [35,36,37]. The spatial datasets for these stressors were compiled accordingly (Table 3). The impact of each stressor on the habitat was scored from the literature review, on a scale of 1 to 3 based on the exposure and consequence criteria (Table 4).

HRA model analysis was performed using InVEST (version 3.14.2) (https://naturalcapitalproject.stanford.edu, accessed on 14 August 2024) with a Euclidean risk calculation and a linear decay function to represent the decreasing influence of stressors with distance. All outputs were generated at a spatial resolution of 30 m × 30 m. Additionally, the HRA model offers a method for identifying high-risk areas by categorizing the risk from stressors into three levels in each grid cells of the habitat [32]. To identify high-risk areas within suitable habitats, the risk scores were normalized to the maximum value and classified into three categories: low risk (0–33%), medium risk (33–66%), and high risk (66–100%).

### 2.6. Spatial Prioritization and Core Conservation Areas

Zonation is a decision-support tool used to identify areas of high conservation value based on biodiversity features and spatial distributions, thereby supporting the expansion or designation of PAs [69]. The algorithm evaluates the entire landscape under the assumption that all spatial units are potential conservation targets, iteratively removing cells with the lowest conservation value (0) according to user-defined input settings [70]. Spatial prioritization follows the principle of marginal loss, applying a cell-removal rule to minimize the loss of conservation value across the landscape. Two primary cell-removal rules are available: Additive Benefit Function (ABF) and Core-Area Zonation (CAZ), each emphasizing different aspects of conservation value [71,72].

In this study, Zonation 5 (version 2.1) (https://zonationteam.github.io/Zonation5, accessed on 6 March 2024) was used to integrate the outputs from the MaxEnt and HRA models. Potential habitat from the MaxEnt model served as the feature layer, and the suitable area was applied as a hierarchical mask layer to assign the highest conservation value. High-risk areas identified by the HRA model were used as a condition layer and excluded from the conservation prioritization process. The CAZ rule was selected to emphasize the protection of core areas. The results were classified into five ranks using natural breaks in ArcGIS Pro, with the top-ranked zones designated as High-Priority Areas (HPAs).

To evaluate whether these HPAs correspond to areas of recognized ecological importance, a spatial overlay analysis was conducted using the 1st-grade areas from the Ecological and Nature Map (ENM). The ENM classifies natural environments into grades ecological value, naturalness, and landscape value based on 14 types of ecological surveys considering [73,74]. The four main criteria are vegetation, endangered wildlife, wetlands, and topography, and areas can be classified into 1st-grade, 2nd-grade, 3rd-grade, and special management areas [73]. The 1st-grade areas are considered ecologically critical habitats for endangered species, warranting priority protection and restoration [75]. The overlay results identified conservation areas for the long-tailed goral that overlap with ecologically valuable areas, as well as HPAs outside these zones, thereby highlighting additional areas in need of protection. The spatial data for the ENM were obtained from the Environmental Geographic Information Service (https://egis.me.go.kr, accessed on 1 March 2025), with a total study area of 968.86 km^2^ (Figure 4).

## 3. Results and Discussion

### 3.1. Potential Habitat for the Long-Tailed Goral

The MaxEnt model analysis for the long-tailed goral achieved an AUC value of 0.866, indicating high predictive accuracy, and the MTSS threshold was identified as 0.323. Among the environmental variables, distance from agricultural areas had the most significant influence on model performance, accounting for 25.5% of the variance. This was followed by distance from mountain trails (18.6%), forest age class (10.9%), distance from water bodies (9.4%), and elevation (7.7%). The highest occurrence probabilities of the long-tailed goral were observed at elevations between 300 and 500 m. The occurrence probability increased with proximity to water bodies, whereas distances greater than 2000 m from mountain trails and agricultural areas were associated with a higher likelihood of distribution. Among vegetation-related factors, forest age of class 4 and class 6 were associated with increased goral presence. Although research on the relationship between agricultural areas and the distribution of the long-tailed goral is limited, previous studies have reported that the probability of occurrence increases with greater distance from agricultural areas [30]. Water and vegetation serve as essential resources and habitats for wildlife, with the occurrence probability increasing as proximity to these features decreases [8,34]. In contrast, mountain trails, which fragment the habitat of the long-tailed goral and generate continuous disturbances, exhibited a positive relationship between distance and occurrence, indicating a higher presence in areas farther from the mountain trails [26,27]. Elevation was also identified as a key environmental determinant of goral distribution, indicating the finding of this study appropriately accounted for the elevation characteristics relevant to the species’ habitat preferences [8]. However, the importance of forest habitats indicated in the results may be inflated due to site selection bias, as the study areas were predominantly chosen from regions with high forest cover and relatively dense goral populations. Nevertheless, this study highlights the importance of mid-elevation habitats for the long-tailed goral. These areas provide suitable forest conditions and serve as ecological corridors, and should be prioritized in conservation planning. Such areas could also be considered as buffer zones for existing protected areas or as candidate sites for new designations.

Figure 5a shows the potential habitat predicted by the MaxEnt model. The model projected a high probability of the long-tailed goral occurrence in the forested regions of Uljin, consistent with previous reports of stable the long-tailed goral populations in the Uljin, Samcheok, and Bonghwa regions [8]. Using the MTSS threshold value of 0.323, areas with a high probability of occurrence were extracted from the potential habitat map. The total suitable area was estimated at 467.52 km^2^, accounting for 10.39% of the total study area. This area was primarily concentrated in the forested regions of Uljin, Gyeongbuk Province (Figure 5b). However, the MaxEnt predictions may not fully represent the actual distribution patterns of the long-tailed goral because most occurrence records are obtained from accessible locations, and the long-tailed goral typically inhabit rugged terrain with limited human access [76]. The MaxEnt model does not consider habitat quality or condition. Therefore, integrating it with models such as the InVEST Habitat Quality model could improve predictive accuracy, enabling habitat suitability assessments that account for both potential distribution and multiple interacting environmental factors. In South Korea, however, stressors such as hunting or livestock grazing are not common in the mountainous habitats of the long-tailed goral due to strict legal protection and limited pastoral activities. Therefore, these factors were not considered in this study, in addition to the lack of spatially explicit data.

### 3.2. Risk Areas in the Habitat of the Long-Tailed Goral

The results of the HRA model for the habitat of the long-tailed goral indicated an average risk value of 0.304 for the five stressors. Among these stressors, roads were identified as the most significant contributor to habitat risk (Table 5). Roads cause habitat destruction and fragmentation, while their construction and operation increase wildlife mortality through disturbances, noise, and vehicle collisions [39,41,43]. While the long-tailed goral generally occupies small home ranges, approximately 1 km^2^ [27,34], and rarely ventures beyond its habitats, resulting in lower road-related risks compared to other mammals that frequently encounter vehicle collisions, roads remain a critical threat to wildlife [42]. The extensive road network within the study area likely played a significant role in the elevated risk values observed.

The cumulative habitat risk values ranged from 0 to 11.141, with the highest recorded in Taebaek, Gangwon Province (Figure 6a). This region is characterized by a combination of major stressors, including roads, mountain trails, deforestation, quarries, and large-scale terrestrial wind power plants. The cumulative impact of these stressors, particularly when located within forest interiors and along ridgelines, can cause long-term ecological disturbances.

Based on risk classification, high-risk areas accounted for 179.54 km^2^, medium-risk areas for 365.23 km^2^, and low-risk areas for 245.55 km^2^. Using a high-risk threshold value of 1.64, the identified high-risk areas were distributed throughout the habitat where multiple stressors overlapped, particularly around road and mountain trail networks (Figure 6b).

### 3.3. Prioritizing Conservation Areas

The spatial prioritization analysis using Zonation assigned the highest conservation priority scores to areas concentrated in Samcheok, Uljin, and Bonghwa, located along the border between Gangwon and North Gyeongbuk Provinces (Figure 7). This distribution is consistent with the high habitat suitability values identified in the MaxEnt model results. In contrast, regions classified as high-risk areas in the HRA model received comparatively lower conservation priority scores.

The spatial distribution of conservation priorities indicated that the lowest priority rank covered 881.63 km^2^ (19.6%), whereas the highest priority rank encompassed 916.91 km^2^ (20.4%) (Table 6). The highest-ranked areas, defined as High-priority Areas (HPAs), were most extensively concentrated in Uljin, which accounted for 351.25 km^2^ of the total HPA extent (Table 6). This finding is consistent with previous studies reporting stable populations of the long-tailed goral in the Uljin and Samcheok regions [8,36], highlighting the importance of prioritizing conservation efforts in Uljin as a key distribution area for the species.

### 3.4. Ecological Value of the Core Conservation Areas

To evaluate the ecological importance of the Zonation results, the HPAs were overlaid with the 1st-grade area of ENM to identify Core Conservation Areas (CCAs). The overlap analysis revealed that 291.721 km^2^ (31.8%) of the HPAs coincided with ENM 1st-grade areas, highlighted in yellow in Figure 8. The study sites encompass the Baekdudaegan Mountains, ranging from Taebaek to Bonghwa, and are characterized by high forest cover with substantial vegetation and landscape value. The 1st-grade area covered approximately 21.5% of the total area, likely reflecting the influence of the forested landscapes in the study site. As effective protected area management should incorporate buffer zones surrounding these core areas [77,78], conservation planning must prioritize CCAs as primary protection targets while also considering HPAs located outside the ENM 1st-grade areas when revising or expanding protected area boundaries.

In this study, relative values were assigned to input data during the prioritization analysis to identify conservation areas. However, incorporating additional factors, such as habitat connectivity, into Zonation analyses could enable a more comprehensive assessment. Furthermore, although core areas were identified through overlap analysis with 1st-grade Ecology and Nature Map regions, future studies should extend beyond delineating spatial boundaries to include evaluations of the actual ecological integrity of both high-priority and core conservation areas. These findings provide a scientific basis for future conservation policies for the long-tailed goral and can serve as baseline data for expanding protected areas centered on HPAs while integrating additional ecological value layers such as ENM classifications.

## 4. Conclusions

This study prioritized habitats for the conservation of the long-tailed goral (*Naemorhedus caudatus*) in South Korea by integrating potential habitat suitability with key anthropogenic threats. A sequential approach was implemented comprising: (1) distribution prediction using the MaxEnt model, (2) habitat risk assessment via the InVEST HRA model, (3) spatial prioritization through Zonation analysis, and (4) ecological validation by overlaying HPAs with 1st-grade areas from the ENM to delineate CCAs.

This study provides an independent, nationwide habitat prediction reflecting the ecological characteristics of the species, offering a foundational reference for effective conservation planning. As the 6th Nationwide Natural Environmental Survey (2024–2028) progresses, incorporating additional occurrence data is expected to improve model accuracy and enable updated nationwide predictions.

Although flexible in defining risk criteria, the HRA model has had limited terrestrial applications, with most previous studies targeting marine ecosystems. Habitat risk was assessed using five major threat factors; however, the absence of quantitative analysis on interactions and relative influence among stressors, coupled with reliance on prior literature for evaluation criteria, represents a notable limitation. Future research should incorporate empirical data from actual habitats, conduct quantitative validation of model outputs, and perform field-based verification of high-risk areas to enhance the robustness and policy applicability of the results.

Zonation prioritization proved effective for spatial conservation planning, highlighting the need to incorporate previously unprotected HPAs identified through ENM overlap analysis. While the current prioritization was based on the relative value of input data layers, integrating additional parameters, such as habitat connectivity, could provide a more comprehensive framework for conservation. Furthermore, assessing the ecological integrity of both HPAs and CCAs would strengthen the scientific basis for decisions related to the designation or revision of PAs within national land use planning.

Overall, this study presents a methodological framework that integrates habitat prediction, risk assessment, and spatial prioritization to identify conservation priorities for the endangered long-tailed goral. The findings deliver policy-relevant insights and offer a scientific foundation for the designation, management, and adaptive expansion of protected areas, particularly when combined with field investigations and follow-up research.

## Figures and Tables

**Figure 1 animals-15-02848-f001:**
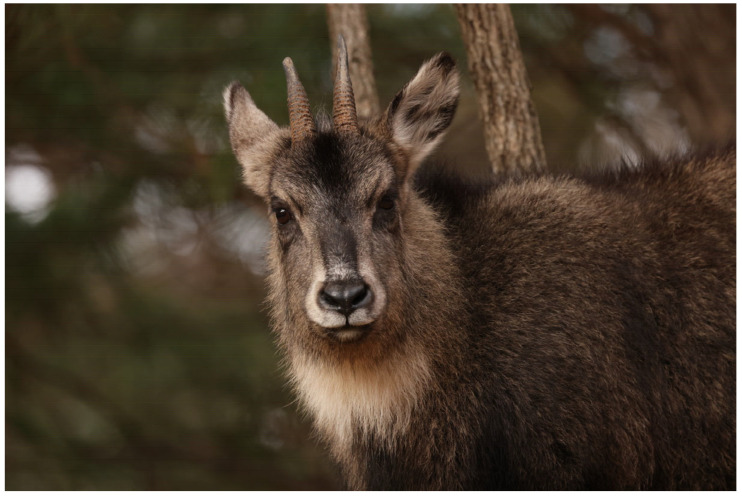
The appearance of the long-tailed goral (*Naemorhedus caudatus*) [11].

**Figure 2 animals-15-02848-f002:**
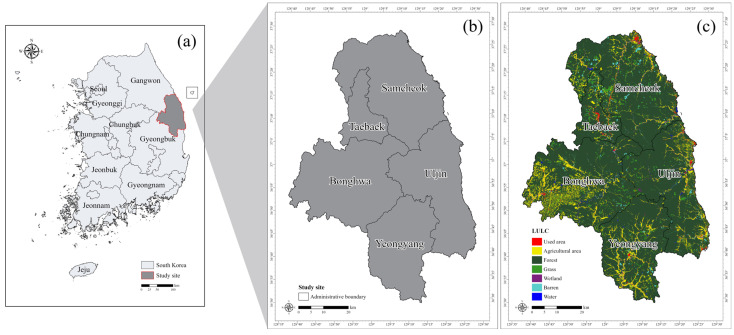
(**a**) Location of the study site in South Korea, (**b**) administrative boundaries of the five provincial areas, and (**c**) land use distribution within the study site.

**Figure 3 animals-15-02848-f003:**
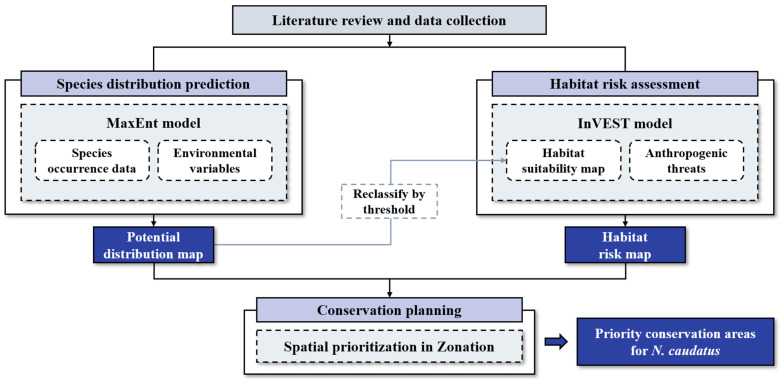
Methodological framework employed in this study.

**Figure 4 animals-15-02848-f004:**
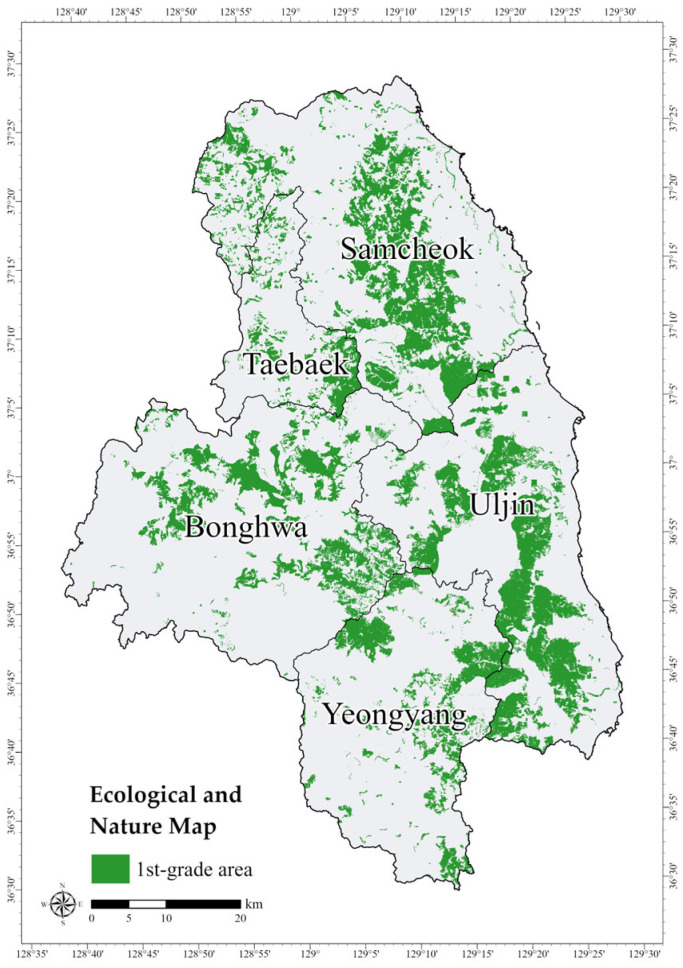
Distribution of 1st-grade areas in the Ecological and Nature Map.

**Figure 5 animals-15-02848-f005:**
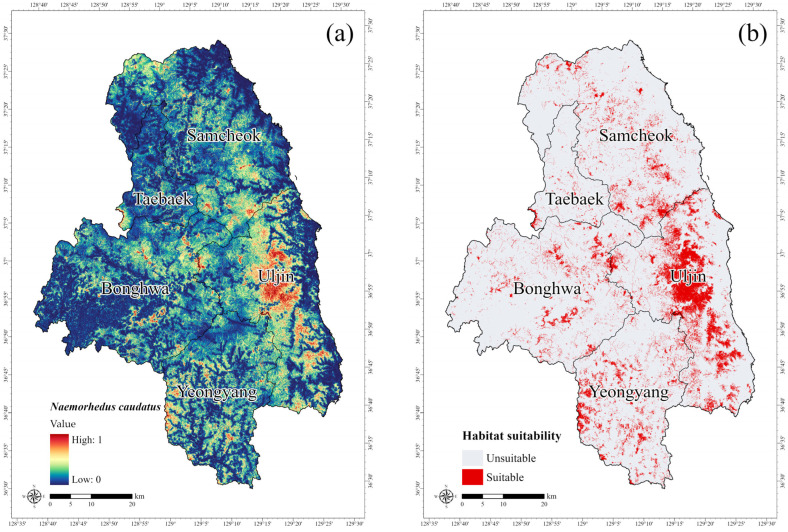
(**a**) Potential distribution map of the long-tailed goral (*Naemorhedus caudatus*), and (**b**) suitable and unsuitable areas classified by the MTSS threshold. The areas shaded in red indicate a higher probability of occurrence.

**Figure 6 animals-15-02848-f006:**
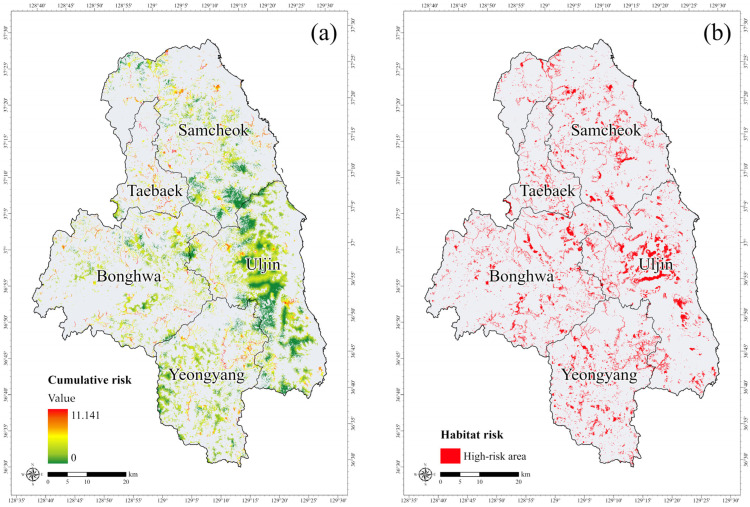
(**a**) Habitat risk map of the long-tailed goral (*Naemorhedus caudatus*) within suitable areas, and (**b**) delineation of high-risk areas based on the habitat risk map.

**Figure 7 animals-15-02848-f007:**
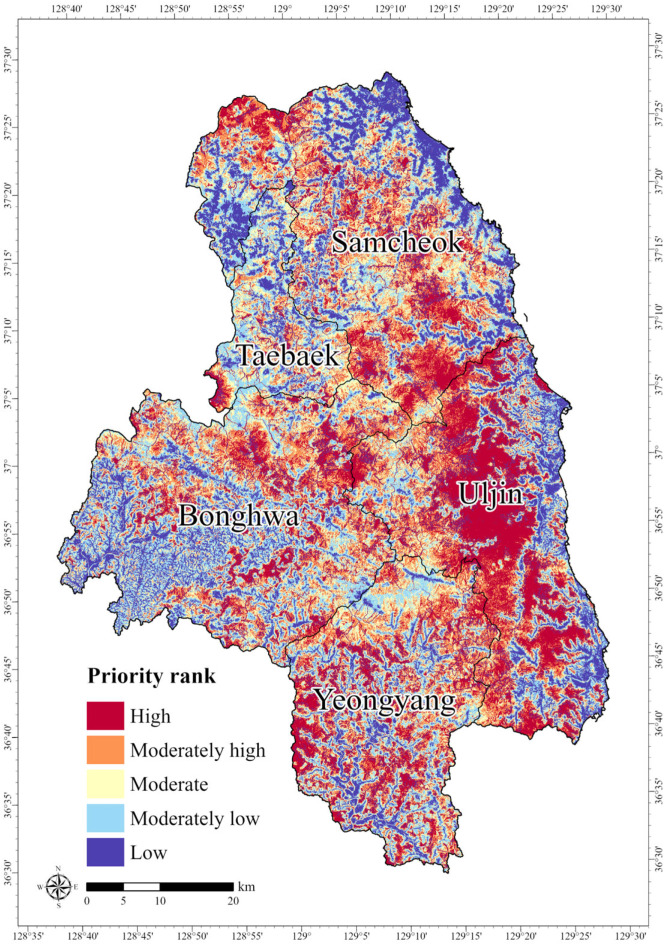
Spatial distribution of prioritized conservation areas for the long-tailed goral (*Naemorhedus caudatus*).

**Figure 8 animals-15-02848-f008:**
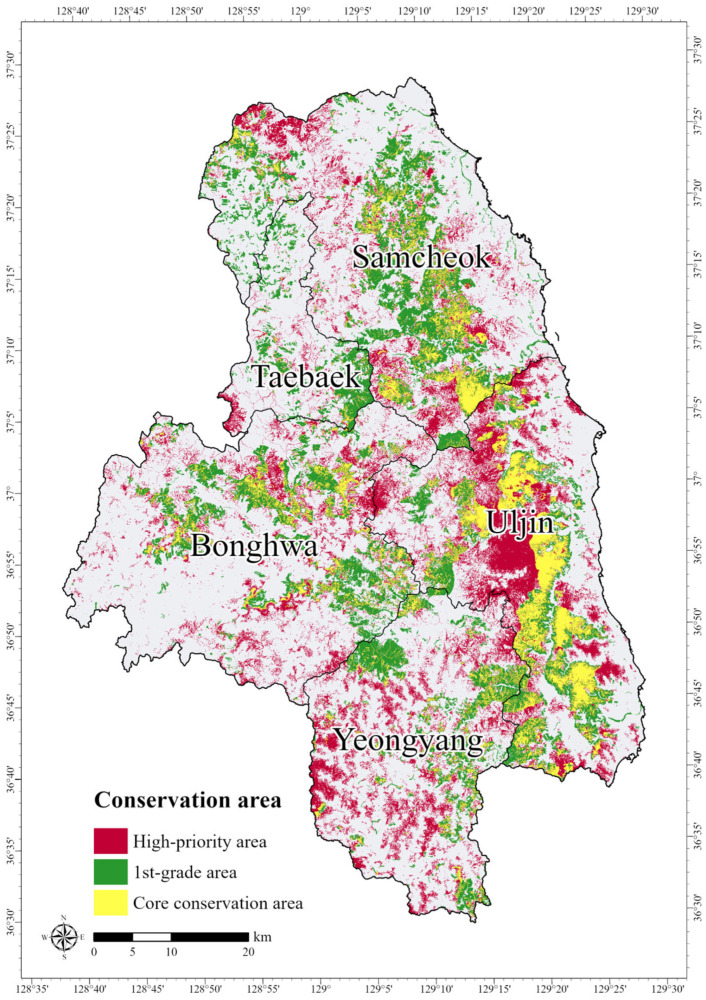
Core conservation areas for the long-tailed goral (*Naemorhedus caudatus*) identified through overlap analysis. Red shading indicates high conservation priority areas, green indicates areas of high ecological value, and yellow represents the overlapping core conservation areas.

**Table 1 animals-15-02848-t001:** Description of the 11 environmental variables used in the MaxEnt model.

Classification	Variable Code	Description	Data Type	Data Source
Topography	DEM	Elevation	Continuous	National Geographic Information Institute (2023) (https://map.ngii.go.kr, accessed on 19 March 2025)
	SLOPE	Gradient		
	ASPECT	Aspect		
Distance	DFW	Distance from water bodies		Environmental Geographic Information Service (2022) (https://egis.me.go.kr, accessed on 10 March 2025)
	DFA	Distance from agricultural areas		
	DFT	Distance from traffic areas		
	DFM	Distance from mountain trails		Korea Forest Service (2023) (https://map.forest.go.kr, accessed on 31 March 2025)
Vegetation	NDVI	Normalized difference vegetation index		Korea Institute of Geoscience and Mineral Resources (2022) (https://www.bigdata-environment.kr, accessed on 7 October 2024)
	FRTP	Forest type class	Categorical	Korea Forest Service (2023) (https://map.forest.go.kr, accessed on 4 October 2024)
	AGCLS	Forest age class		
Land cover	LULC	Land use and land cover		Environmental Geographic Information Service (2022) (https://egis.me.go.kr, accessed on 10 March 2025)

**Table 2 animals-15-02848-t002:** Habitat resilience ratings in the HRA model for the habitat of the long-tailed goral (*Naemorhedus caudatus*) [32].

Habitat Resilience Attributes	Criteria Type	Rating	Data Quality	Weight
Recruitment rate	Consequence	3	1	2
Natural mortality rate		2	1	2
Connectivity rate		3	1	2
Recovery time		2	1	2

**Table 3 animals-15-02848-t003:** Description of the five stressors used in the HRA model for the habitat of the long-tailed goral (*Naemorhedus caudatus*).

Stressor Code	Description	Data Source
ROAD	Roads	Environmental Geographic Information Service (2022) (https://egis.me.go.kr, accessed on 21 April 2025)
DEFOR	Deforestation	
QUAR	Quarries	
MTT	Mountain trails	Korea Forest Service (2023) (https://map.forest.go.kr, accessed on 31 March 2025)
WPP	Wind power plants	Korea Energy Agency (2022) (https://nr.energy.or.kr, accessed on 28 March 2025)

**Table 4 animals-15-02848-t004:** Rating of the five stressors used in the HRA model for the habitat of the long-tailed goral (*Naemorhedus caudatus*), based on criteria, data quality, and weight. Abbreviations: TOR, Temporal overlap rating; MER, Management effectiveness rating; IR, Intensity rating; FDR, Frequency of disturbance rating; CAR, Change in area rating; CSR, Change in structure rating [32].

Stressor	Criteria Type		Rating	Data Quality	Weight	References
ROAD	Exposure	TOR	3	2	1	[27,35,36,38,39,40,41,42,43]
		MER	2	2	3	
		IR	2	2	1	
	Consequence	FDR	3	2	3	
		CAR	2	2	1	
		CSR	3	2	1	
MTT	Exposure	TOR	2	2	2	[35,36,44,45,46,47]
		MER	3	2	2	
		IR	3	2	1	
	Consequence	FDR	3	2	3	
		CAR	2	2	2	
		CSR	3	2	1	
DEFOR	Exposure	TOR	3	2	3	[36,38,48,49,50,51,52,53,54,55,56]
		MER	3	2	3	
		IR	2	2	2	
	Consequence	FDR	2	2	2	
		CAR	3	2	1	
		CSR	3	2	1	
QUAR	Exposure	TOR	3	2	2	[36,57,58,59,60]
		MER	2	2	3	
		IR	3	2	1	
	Consequence	FDR	3	2	3	
		CAR	3	2	1	
		CSR	3	2	1	
WPP	Exposure	TOR	3	2	3	[37,61,62,63,64,65,66,67,68]
		MER	3	2	3	
		IR	2	2	2	
	Consequence	FDR	1	2	3	
		CAR	3	2	3	
		CSR	3	2	1	

**Table 5 animals-15-02848-t005:** Average exposure, consequence, and risk values for the five stressors in the HRA model results.

Stressor Code	Description	Exposure	Consequence	Risk
ROAD	Roads	1.685	1.752	1.122
MTT	Mountain trails	0.391	0.365	0.266
DEFOR	Deforestation	0.131	1.134	0.072
QUAR	Quarries	0.090	0.086	0.047
WPP	Wind power plants	0.027	0.027	0.015
Average value		0.465	0.473	0.304

**Table 6 animals-15-02848-t006:** Prioritized conservation areas for the long-tailed goral (*Naemorhedus caudatus*) across administrative regions (unit: km^2^).

Province	Priority Rank				
Classification	Low	Moderately low	Moderate	Moderately High	High
Uljin	185.9	138.28	132.98	183.69	351.25
Samcheok	269.38	215.86	251.96	249.37	198.18
Yeongyang	134.14	154.64	174.52	181.69	172.54
Bonghwa	232.68	301.29	257.05	236.13	172.23
Taebaek	59.53	89.21	82.76	48.39	22.71
Total	881.63	899.28	899.27	899.27	916.91

## Data Availability

Original dataset will be provided upon request.

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
