# Peer review of "Integrating Habitat Prediction and Risk Assessment to Prioritize Conservation Areas for the Long-Tailed Goral (Naemorhedus caudatus)"

_animals, 2025, doi:10.3390/ani15192848_

Round 1

Reviewer 1 Report

Comments and Suggestions for Authors

The manuscript by Park et al., Integrating Habitat Prediction and Risk Assessment to Prioritize Conservation Areas for the Long-tailed Goral (Naemorhedus caudatus), presents a timely and important contribution to systematic conservation planning for a nationally endangered species in South Korea. By integrating MaxEnt habitat modeling, InVEST-based risk assessment, and Zonation prioritization, the study demonstrates a comprehensive methodological framework. The manuscript is interesting to the readers of Animals and holds a merit for publication. However, several conceptual and methodological issues need attention before the work can be considered for publication. I have listed some major issues here and provided detailed comments in the annotated PDF. 

  • Abstract: It is better to explain what variables were used and what issue was examined rather than writing which software or tools were used. 
  • The introduction establishes the conservation importance of the long-tailed goral, but some claims are overstated or imprecise. For instance, the statement that MaxEnt is specifically suited for the goral is misleading, as the model has wide applicability across taxa. MaxEnt is used not only for long-tailed goral. This sentence is misleading. I suggest the authors to revise it and discuss the application of MaxEnt in different purposes. 
  • The description of the research gap is insufficient; the claim of limited application of these models for the goral should be explicitly distinguished from prior studies.
  • Please use common name in the entire text except mentioning the scientific name at the first appearance. The manuscript has used common and scientific name randomly throughout. 
  • The materials and methods section is generally detailed but lacks critical clarity. Line 98-100: Isn't this selection criteria biased? Does the conservation need in only areas having high population density of goral?Site selection appears biased toward areas with higher population densities and high forest cover, potentially inflating the apparent importance of forest habitats in the results. This bias undermines the generalizability of the findings and requires explicit acknowledgment.
  • Line 106: You have already selected the areas of high forest coverage as study sites. Consequently, your results also suggest the importance of high forest coverage in protection of gorals. You should think about accounting the bias introduced due to your study site selection scheme. 
  • Line 131: Out of how many total records were these 61 points retrieved? Please mention specifically. 
  • As the authors discuss in results and discussion section, the occurrence points were mostly from accessible areas rather than representing the entire distribution range of the long tailed gorals. This limitation needs to be acknowledged. 
  • Line 155-156: Only five anthropogenic stressors were included in the risk assessment, excluding others such as hunting, livestock grazing, or general human disturbance, which may be equally important for goral survival.
  • Further, how were those stressors scaled for the analysis? It needs clear description for the ease of readers. 
  • Similar to the lack of description about the criteria for scaling stressors,  the selection of thresholds for classifying habitat suitability and risk categories are not sufficiently explained. These methodological gaps limit the transparency and reproducibility of the work.
  • The results are well-structured but occasionally superficial. For example, the presentation of variable contributions to the MaxEnt model could be condensed, with greater emphasis placed on ecological interpretation such as what ranges of forest age or elevation are optimal for gorals (lines 196-198).
  • The discussion underplays a major limitation: the reliance on occurrence records from accessible areas violates a core assumption of species distribution models and may distort habitat predictions. This issue should be addressed more thoroughly, perhaps through sensitivity analyses or additional validation.
  • The conclusion is overly long, reiterating results rather than synthesizing key conservation insights. Much of the content fits better in the introduction or discussion. A concise conclusion highlighting the study’s main contributions, limitations, and policy relevance would be more effective.

Minor issues:

  • Tables are inserted as figures/shapes. So, not editable.
  • At many places use of terms like vague- such as "small home ranges", "multiple stressors concentrated areas". Please mention the values or areas specifically. 

Please see the details in annotated PDF. I hope it will be helpful to improve the manuscript. 

Author Response

Please find the attached file for reviewer #1 comments. Thank you for your comments for the manuscript

Reviewer 2 Report

Comments and Suggestions for Authors

This manuscript describes the use of computer models to prioritize conservation areas for the endangered long-tailed goral (Naemorhedus caudatus) in South Korea , which are threatened by a variety of anthropogenetic  factors.  The presentation is clearly written, with attractive illustrations. The topic is a good fit for the journal, and my comments are limited to a few small issues. My biggest suggestion for change is to eliminate the appendix and incorporate this information within the main body of the text.

Line 88: N. caudatus is a bovid mammal of the order Artiodactyla that retains many primitive characteristics of ancestral bovids.

Comment: I suggest that you should briefly explain what you mean by primitive characteristics of ancestral bovids.

Also, please include an image of what these animals look like.

Figure 1b,c: The text showing the names of provincial areas is not legible. It would help to replace the highlighted fonts with simple fonts.

In the methods section, please explain the sources for the software used for the RStudio, MaxInt and InVest models.

Table 2: I suggest that the abbreviations for stressors should be explained in the caption. In two places in the text (lines 155 and 234) provide an explanation: Five primary stressors were identified: roads, mountain trails, deforestation, quarries, and wind power plants. The line 234 explanation would work better if it were included as part of the Table 2 caption.

Figure 4b: the red square that is labeled as “high risk area” is not a numerical value, and “value” heading  can be deleted. In contrast, Figure 4b shows  a range of numerical values, so the “value” heading is appropriate.

The information presented in the Appendix adds confusion to the general organization. I suggest that this information can be incorporated into the main text of the manuscript. The overall word count would remain the same, and the length of the manuscript is short enough that there is no need to move these data to an accessory file.

Author Response

Please find the attched file for the reply to the reviewer #2. Thank you for your comments and we believe that our article improved much better thanks to your comments.

Round 2

Reviewer 1 Report

Comments and Suggestions for Authors

Dear authors,

Thank you for incorporating the comments and suggestions and also making point-by-point responses to the review comments. 

All the best!